# Large carbon sink potential of secondary forests in the Brazilian Amazon to mitigate climate change

Viola H. A. Heinrich [1✉], Ricardo Dalagnol[2], Henrique L. G. Cassol [2], Thais M. Rosan[3], Catherine Torres de Almeida[2], Celso H. L. Silva Junior [2], Wesley A. Campanharo [2], Joanna I. House [1,4], Stephen Sitch [3], Tristram C. Hales[5], Marcos Adami [6], Liana O. Anderson [7] & Luiz E. O. C. Aragão[2,3]

Tropical secondary forests sequester carbon up to 20 times faster than old-growth forests. This rate does not capture spatial regrowth patterns due to environmental and disturbance drivers. Here we quantify the influence of such drivers on the rate and spatial patterns of regrowth in the Brazilian Amazon using satellite data. Carbon sequestration rates of young secondary forests (<20 years) in the west are ~60% higher ($3.0 \pm 1.0$ Mg C ha$^{-1}$ yr$^{-1}$) compared to those in the east ($1.3 \pm 0.3$ Mg C ha$^{-1}$ yr$^{-1}$). Disturbances reduce regrowth rates by 8–55%. The 2017 secondary forest carbon stock, of 294 Tg C, could be 8% higher by avoiding fires and repeated deforestation. Maintaining the 2017 secondary forest area has the potential to accumulate ~19.0 Tg C yr$^{-1}$ until 2030, contributing ~5.5% to Brazil's 2030 net emissions reduction target. Implementing legal mechanisms to protect and expand secondary forests whilst supporting old-growth conservation is, therefore, key to realising their potential as a nature-based climate solution.

[1] School of Geographical Sciences, University of Bristol, Bristol, UK. [2] Earth Observation and Geoinformatics Division, National Institute for Space Research (INPE), São José dos Campos, Brazil. [3] College of Life and Environmental Sciences, University of Exeter, Exeter, UK. [4] Cabot institute, University of Bristol, Bristol, UK. [5] School of Earth and Environmental Sciences, Cardiff University, Cardiff, UK. [6] Amazon Regional Center, National Institute for Space Research (INPE), Belém, Brazil. [7] National Center for Monitoring and Early Warning of Natural Disaster (CEMADEN), São José dos Campos, Brazil
. ✉email: viola.heinrich@bristol.ac.uk

Global forests are expected to contribute a quarter of the pledged mitigation under the 2015 Paris Agreement, by limiting deforestation and by encouraging forest regrowth[1]. The Brazilian Amazon biome (Amazonia) is the largest continuous tropical forest on Earth, occupying 3% of terrestrial land. It stores ~10% of the global forest carbon (120,000 Tg C)[2,3] and between 2000 and 2010 sequestered ~150 Tg C yr$^{-1}$ through natural growth (5% of global land sink), while emitting ~143 ± 56 Tg C yr$^{-1}$ through deforestation (~1.4% of global carbon emissions)[4–6]. As part of their Nationally Determined Contributions (NDC) to the Paris Agreement, Brazil has pledged to restore and reforest 12 million hectares of forests by 2030 to contribute to net emission reductions[7]. Part of this reduction can be achieved by the natural regeneration of secondary forests on abandoned land, which are already regrowing on ~20% of deforested land in Amazonia[8–10].

Previous estimates of average net carbon uptake in young (<20 years old) secondary forest range between 2.95 ± 0.4 and 3.05 ± 0.5 Mg C ha$^{-1}$ yr$^{-1}$, 11–20 times larger than old-growth primary forests[11,12]. These estimates, which are based on limited field data across the Neotropics, are unable to capture the different spatial patterns and rates of secondary forest carbon sequestration, which are influenced by several drivers. This includes environmental drivers such as shortwave radiation, precipitation, soil fertility and forest water deficit, as well as anthropogenic disturbances like fire and repeated deforestation cycles prior to regrowth[11,13–16]. The secondary forest carbon stock of regions with very high-water deficit (−1200 mm yr$^{-1}$) can be up to 85% lower compared to no water deficit (0 mm yr$^{-1}$) regions in the Neotropics[11]. The effects of these drivers are neither limited to secondary forest growth, nor are they static over space and time, affecting the magnitude of forest carbon sequestration and stocks[17]. A recent study showed that rising annual mean temperatures and drought reduced tree growth in Amazonian old-growth forests[4]. This effect, coupled with ongoing deforestation suggests that the sink in these forests peaked in the 1990s and is now steadily declining[4]. Considering these changes, it is important to obtain a wider spatial and temporal understanding of drivers affecting the magnitude and sustainability of secondary forest regrowth.

Remote sensing products can be used to study these effects, offering broad spatial and temporal coverage. With the availability of nearly four decades of Landsat data (30 m spatial resolution), it is now possible to track the fate of deforested areas over time, which includes the changing demography of secondary forests across Amazonia[10,18]. According to satellite-based analysis, secondary forests are typically part of a 5–10 year cycle of clearance and abandonment since they are currently not protected by national policies aimed at curbing deforestation[19,20]. These repeated deforestations are expected to decrease the carbon sink of future regrowth forests. Deforestation of secondary forests amounted to ~70% of total Amazonian forest loss between 2008 and 2014[21]. However, the relationship between secondary forest regrowth and environmental and disturbance drivers has never been explored spatially explicitly using global remote sensing products.

The primary aim of this study is to provide key advances in understanding the spatial variation of secondary forest regrowth in the Brazilian Amazon, a large and geographically complex region. Previous studies have already provided the first steps to understanding regrowth on a biome scale, influenced by some driving variables[11,14,22]. Here we introduce additional environmental and anthropogenic disturbance drivers that affect regrowth, including local-scale drivers, and for the first time, disaggregate their effects using a spatially explicit approach[11,14,22]. We map secondary forests annually from 1985 to 2017, determine their ages[10,18], and provide the first applications of these maps to analyse secondary forest regrowth in terms of Aboveground Carbon (AGC)[23–25]. We present a map of Amazonian secondary forest regrowth rates with the quantification of the contemporary secondary forest carbon sink considering the impact of different drivers on AGC accumulation. We use this to model the future carbon sequestration potential of secondary forests relative to the Brazilian NDC targets.

We find that secondary forest regrowth and associated carbon sequestration varies across regions within Amazonia, due to both large-scale environmental drivers such as shortwave radiation and local-scale drivers such as fire. Overall, secondary forests in the North-West regrow up to twice as fast (3.0 ± 1.0 Mg C ha$^{-1}$ yr$^{-1}$) compared to eastern regions (1.3 ± 0.3 Mg C ha$^{-1}$ yr$^{-1}$); however, the impact of disturbances such as fire and repeated deforestations prior to regrowth only reduces the regrowth by 20% in the North-West (2.4 ± 0.8 Mg C ha$^{-1}$ yr$^{-1}$) compared to 55% in the North-East (0.8 ± 0.8 Mg C ha$^{-1}$ yr$^{-1}$). We find that the 2017 area of secondary forest, which occupies only ~4% of Amazonia, can contribute significantly (~5.5%) to Brazil's net emissions reduction targets as stated in their NDC, accumulating ~19.0 Tg C yr$^{-1}$ until 2030 if the current area of secondary forest is maintained (13.8 Mha). However, this value reduces rapidly to less than 1% if only secondary forests older than 20 years are preserved (2.2 Mha). We conclude that preserving the remaining old-growth forest carbon stock and implementing legal mechanisms to protect and expand secondary forest areas are key to realising the potential of secondary forest as climate change mitigation solutions.

## Results

**Impact of drivers on secondary forest regrowth.** We used the land-cover product MapBiomas (Collection 3.1)[25] to identify secondary forests and their ages from 1985 to 2017 and used the European Space Agency Climate Change Initiative (ESA-CCI) Aboveground Biomass product to model the regrowth of secondary forests across Amazonia[23] (Supplementary Notes 1–3; Supplementary Figs. 1–6). Based on these two products, we identified and tested the effects of six key drivers on secondary forest regrowth and AGC accumulation: (1) Average annual shortwave (SW) radiation[26]; (2) Average annual precipitation[27]; (3) Forest water deficit using the Maximum Cumulative Water Deficit index (MCWD)[28,29]; (4) Soil fertility using the Soil Cation Concentration (SCC) as a proxy[30]; (5) Burned area[31] and (6) the number of deforestations since the start of the data in 1985 prior to the most recent regrowth, hereon simply termed "repeated deforestations" (see Methods; Supplementary Table 1). Our analysis reveals significant differences in AGC accumulation in secondary forests considering these different drivers (Fig. 1; Supplementary Tables 2–7). After forest age, SW radiation is the most important variable influencing AGC (Fig. 1g). In areas of very low annual SW radiation (<170 Wm$^{-2}$), the overall regrowth rate is almost three times greater compared to areas of high SW radiation (>187 Wm$^{-2}$), ~3.4 ± 0.6 and ~1.3 ± 0.4 Mg C ha$^{-1}$ yr$^{-1}$, respectively (Fig. 1a, Supplementary Table 7). SCC is the second most important environmental driver (Fig. 1g). However, there is no statistical difference in carbon accumulation under different SCC conditions, furthermore the expected trend, increased carbon accumulation with increased soil fertility, is reversed, probably due to the dominant effect of other environmental drivers, which act on larger-regional scales[32–34] (Fig. 1d; Supplementary Table 4). Areas with very low MCWD (>−180 mm yr$^{-1}$) assimilate almost double the carbon compared to areas with very high MCWD (<−350 mm yr$^{-1}$) in the first 20 years of regrowth (2.7 ± 0.7 Mg C ha$^{-1}$ yr$^{-1}$ and 1.5 ± 0.2 Mg C ha$^{-1}$ yr$^{-1}$, respectively) (Fig. 1b, Supplementary Table 2).

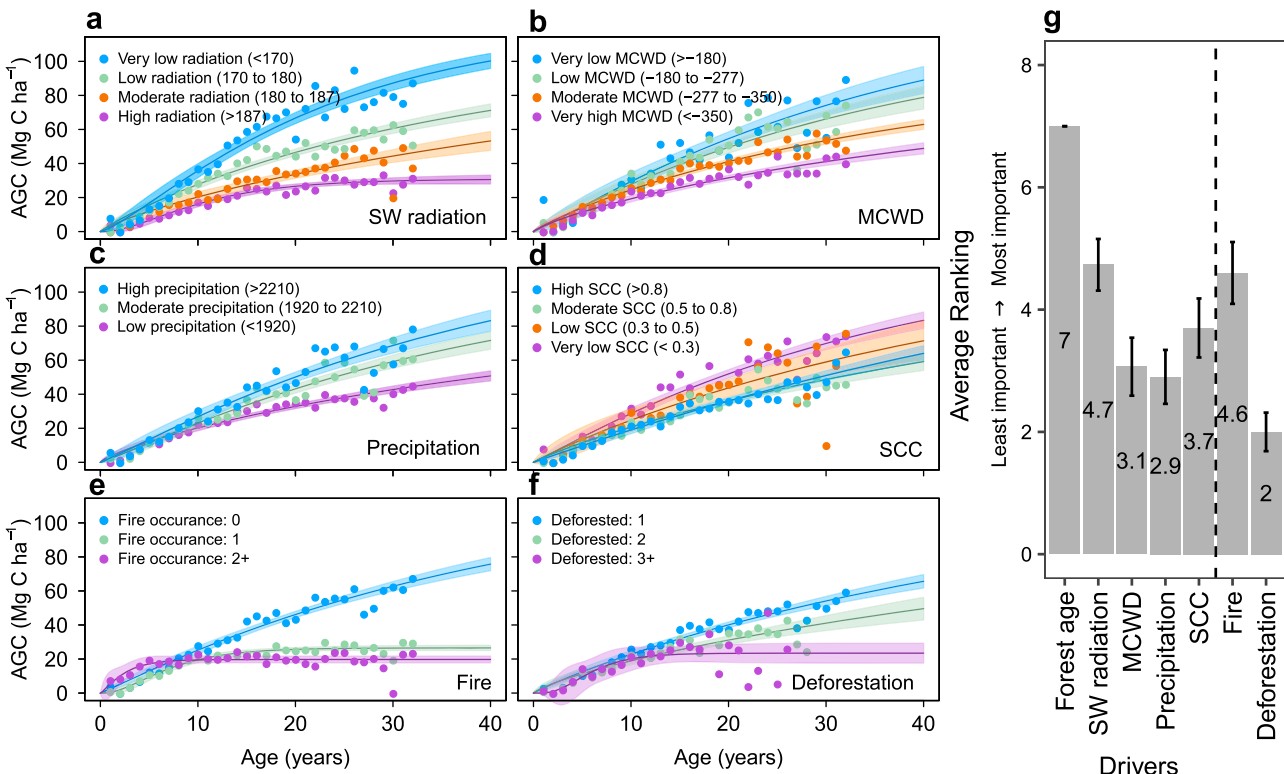

**Fig. 1 Secondary forest carbon accumulation with increasing age under different driving conditions.** Drivers are **a** Annual mean downward shortwave (SW) radiation (Wm$^{-2}$), **b** Maximum Cumulative Water Deficit (MCWD; mm yr$^{-1}$), **c** Annual mean precipitation (mm yr$^{-1}$), **d** Soil Cation Concentration (SCC; cmol($+$) kg$^{-1}$), **e** Fire occurrences between 2001 and 2017, and **f** Number of deforestations prior to regrowth between 1985 and 2017, where 1 refers to areas that have only experienced the original conversion from old-growth forest to secondary forests during the period 1985–2017 with no subsequent deforestation events. The bar graph **g** shows the average importance ranking of the drivers (**a**–**f**), as well as Forest age, in influencing Aboveground Carbon (AGC) accumulation. The average ranking was calculated following 30 iterations of a conditional random-forest model. The importance has been ranked from least important (1) to most important (7) and the vertical dotted line separates environmental drivers (left) from anthropogenic disturbance drivers (right). Shading in **a**–**f** denotes the 95% confidence interval of the models, based on the median value of the initial data for each age—dots in figures. The error bars in **g** denote the 95% confidence interval.

Similar differences in the regrowth rates can be observed under conditions of low mean annual precipitation (<1920 mm yr$^{-1}$) compared to moderate and high conditions (1920–2210 mm yr$^{-1}$ and >2210 mm yr$^{-1}$, respectively) (Fig. 1c, Supplementary Table 6).

For most of our modelled regrowth curves, secondary forests were able to reach AGC levels equivalent to those of old-growth forests; however, the time taken to reach these levels is generally more than a century (Supplementary Table 8). Our results also show that in areas of anthropogenic disturbance such as fires and repeated deforestations, the carbon accumulation rate was up to 75% lower and even plateaued within 11–40 years, thus potentially never recovering to old-growth forest AGC values (Fig. 1e and 1f; Supplementary Table 8). Our results showed that fire occurrence, a predominantly anthropogenic disturbance[35], has a similar importance ranking as the most important environmental driver influencing AGC (Fig. 1g), despite only affecting 29.2% of secondary forest plots (Supplementary Fig. 8b). Repeated deforestations were the least important of the drivers assessed for modelling AGC regrowth across the entire biome (Fig. 1g). However, regrowth rates in secondary forests exposed to 3+ deforestations prior to regrowth were 40% lower compared to secondary forest plots only experiencing one deforestation in the study period (Supplementary Table 8). The overall low importance may be linked to the small number of secondary forest plots being exposed to repeated deforestations (26.3%) (Supplementary Fig. 8a).

**Mapping the spatial patterns of regrowth.** To analyse the spatial variation of regrowth rates in our models, we identified different regions of Amazonia according to the three climate drivers: SW radiation, annual precipitation and MCWD (Fig. 2a). For each spatially explicit climate region we calculated the AGC value of old-growth forest (Supplementary Table 9) and modelled how secondary forest regrowth was further affected by different types of disturbances: fire and deforestation (Fig. 3). Our analysis shows distinct regrowth regimes emerging in these four heterogeneous climate regions (Fig. 3), with regrowth in some regions conditioned largely by natural, environmental drivers, and others by anthropogenic disturbance drivers (Fig. 2b–e). In the North-West, a region with generally high precipitation (mean of 2049 mm yr$^{-1}$), low SW radiation (mean of 163.6 Wm$^{-2}$), little to no water deficit (MCWD mean of −64.4 mm yr$^{-1}$) and low SCC (0.29 cmol ($+$)kg$^{-1}$; Supplementary Fig. 7), regrowth rates were generally the highest and hardly influenced by any kind of disturbance. Here regrowth rates ranged between 2.4 ± 0.9 and 3.0 ± 1.0 Mg C ha$^{-1}$ yr$^{-1}$ in the first 20 years of regrowth (Fig. 3a; Supplementary Table 9). In contrast, the North-East and South-East regions have lower overall regrowth rates (1.3 ± 0.3 and 1.8 ± 0.3 Mg C ha$^{-1}$ yr$^{-1}$, respectively in the first 20 years) with fire and deforestation disturbances reducing their regrowth by around 50% to as low as 0.6 Mg C ha$^{-1}$ yr$^{-1}$ in the North-East in the first 20 years (Fig. 3b–d). In the South-West and South-Eastern regions fire disturbance is, respectively, the first and second most

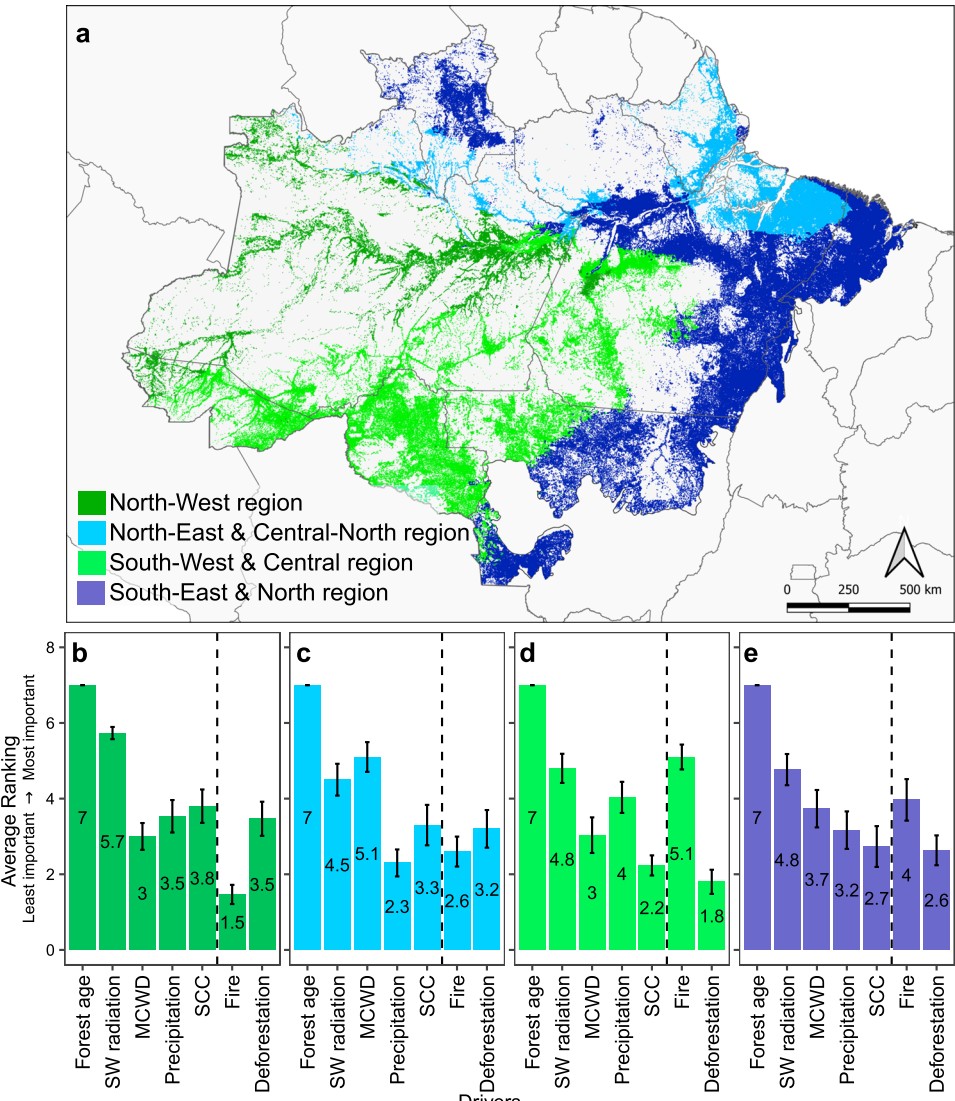

**Fig. 2 Importance ranking of environmental and disturbance drivers on secondary forest regrowth grouped by climatological regions. a** Regions are grouped according to similarities in Maximum Cumulative Water Deficit (MCWD), annual average downward shortwave (SW) radiation and annual average precipitation. See Supplementary Table 9 for quantitative interpretations of the regions. The average importance ranking for each of the six variables, as well as Forest age, is shown for **b** the North-West region, **c** the North-East and Central-North region, **d** the South-West and Central region, and **e** the South-East and North region. The average ranking was calculated following 30 iterations of a conditional random-forest model. The importance has been ranked from least important (1) to most important (7) and the vertical dotted line separates environmental drivers (left) from anthropogenic disturbance drivers (right). The error bars in **b–e** denote the 95% confidence interval. The abbreviation SCC in **b–e** refers to Soil Cation Concentration.

important driver to influence AGC accumulation, after forest age (Fig. 2d, e).

We validated our models with field AGC estimates of secondary forests collected across Amazonia (284 samples across 33 locations) and found that our AGC estimates are statistically similar ($p > 0.01$) within the four regions identified in Fig. 2a (Supplementary Note 4; Supplementary Fig. 10). We also compared the regional models with basin-wide models used in previous studies and within the Brazilian Greenhouse Gas Inventory (GHGI), which do not consider different environmental or anthropogenic disturbance drivers[21,36,37]. In the western regions, during the first 10 years of growth, our models of 'no disturbance' were visually very similar to the other models (Supplementary Fig. 11). We found no significant difference to AGC estimates from the model used in previous research ($p > 0.01$; Supplementary Table 11). Estimates using the equation from the GHGI were significantly higher across the 40 years modelled

in all four regions with disturbed areas having significantly lower regrowth rates ($p < 0.01$; Supplementary Table 11; Supplementary Fig. 11). This highlights the potential importance of being able to disentangle the drivers influencing regrowth in different regions and suggests that the regrowth sink as calculated in the GHGI may be overestimated.

**Modelling the 2017 and future secondary forest sink.** We quantify the net AGC change for the year 2017 by explicitly considering the changes in secondary forest area from 2016 to 2017 and then apply the relevant regrowth model (Fig. 3) based on the four climate regions and the disturbances these forests experienced. Our results show that new regenerating forests and existing secondary forests combined resulted in a carbon sink of 28.0 Tg C yr$^{-1}$, while 16.1 Tg C yr$^{-1}$ was emitted from the reduction in secondary forest area due to deforestation, resulting

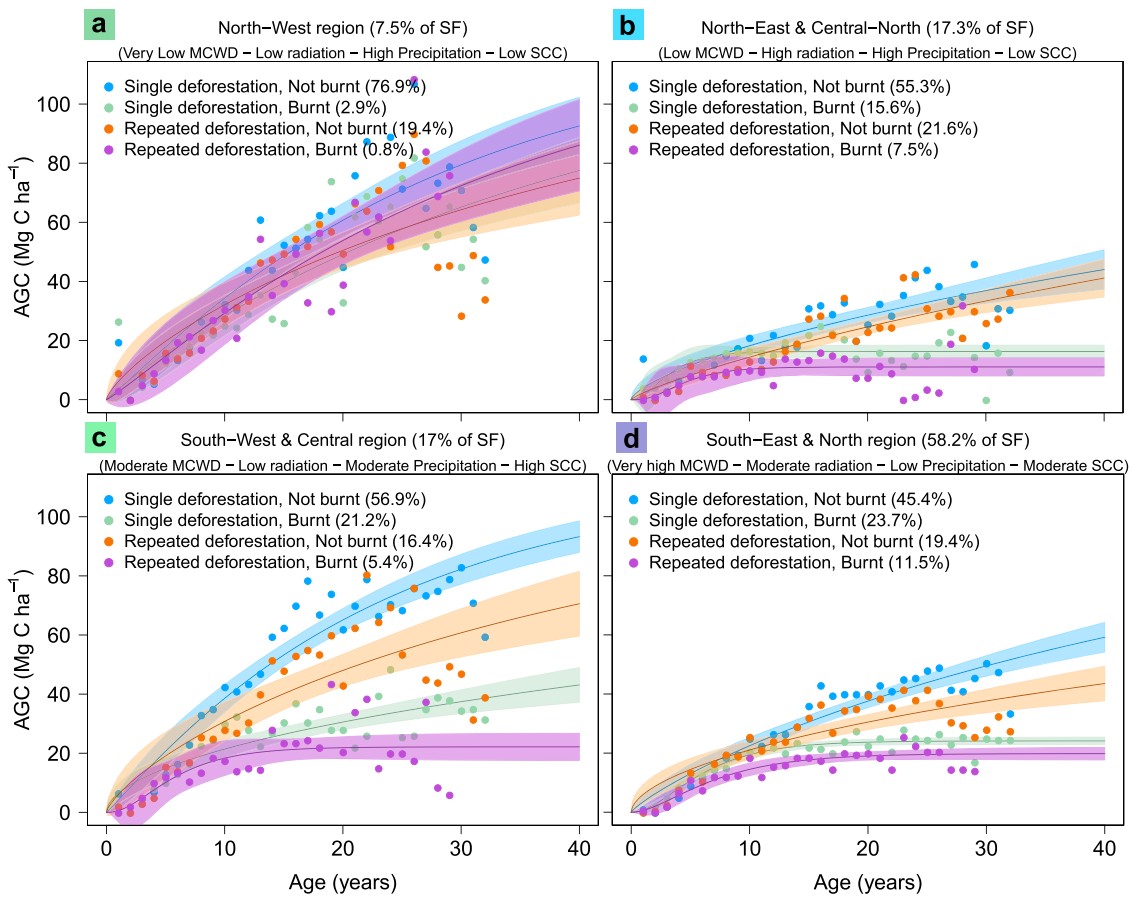

**Fig. 3 The effect of disturbance on region-specific regrowth models of carbon accumulation in secondary forests.** The secondary forest regrowth models are shown for the regions identified in Fig. 2a. In each region, the climatological variables of Maximum Cumulative Water Deficit (MCWD), annual average shortwave radiation and annual average precipitation are: **a** North-West region, **b** North-East and Central-North region, **c** South-West and Central region, **d** South-East and North region. The legend colours represent percentage of secondary forests that are affected by the type of disturbance in each region prior to regrowth, namely: blue—No disturbance ("Single deforestation, Not burnt"), green—Only fire disturbance ("Single deforestation, Burnt"), orange—Only deforestation disturbance ("Repeated deforestation, Not burnt"), purple—Both disturbances ("Repeated deforestation, Burnt). Shading denotes the 95% confidence interval of the models based on the median value of the initial data for each age—dots in figures. See Supplementary Table 9 for quantitative interpretations of the qualitative definitions given here, for example "Low precipitation". The abbreviation SCC in **a**–**d** refers to Soil Cation Concentration.

in a net secondary forest carbon sink of 11.9 Tg C yr⁻¹ (Fig. 4a–c). We find the total carbon stored in all Amazonian secondary forests in 2017 to be 293.7 Tg C (Fig. 4d). We also estimate that the potential carbon stock if all secondary forests had regrown without experiencing any disturbances since the beginning of the study period, could have reached 319.7 Tg C in 2017 (Fig. 4d).

Finally, to quantify the potential of the existing 2017 secondary forests to contribute to reducing future net carbon emissions according to Brazil's NDC, we modelled future potential stocks and the annual carbon sink for the decade ahead by considering various levels of preservation (Fig. 5). Until 2030 we project a 90% difference in carbon accumulation between the most ambitious preservation plan (preserving all 13.8 Mha of secondary forest areas) and the least ambitious plan (preserving 2.2 Mha including only secondary forests older than 20 years in 2017; Fig. 5b), with ~19.0 ± 2.4 Tg C yr⁻¹ and 2.0 ± 0.2 Tg C yr⁻¹ being accumulated for the two plans, respectively.

## Discussion

In this study, we quantified the impact of environmental and anthropogenic disturbance drivers on carbon accumulation in

Amazonian secondary forests. SW radiation was the most important driving variable influencing secondary forest regrowth (Fig. 1), with the areas of lowest SW radiation observed in western Amazonia (~163.6 Wm⁻²) having the highest regrowth rates ranging between 3.0 ± 1.0 and 3.2 ± 0.6 Mg C ha⁻¹ yr⁻¹, considering no disturbances (Supplementary Figs. 7 and 9; Supplementary table 9). These estimates of carbon accumulation in the western regions are similar to previous estimates of 2.95 ± 0.4 and 3.05 ± 0.5 Mg C ha⁻¹ yr⁻¹ [11,12], but this is the first time it has been explored spatially explicitly considering environmental drivers such as SW radiation. The higher estimated regrowth rates in areas of lower SW radiation (Figs. 1 and 3a) are likely linked to higher cloud cover resulting in more diffuse radiation and lower vapour pressure deficit. Diffuse radiation can penetrate deeper into closed forest canopies than direct shortwave radiation and enhance productivity and thereby carbon sequestration[13,38], while a lower vapour pressure deficit encourages leaf stomata to remain open, maximising productivity and thereby regrowth[39].

There are synergies, or spatial correlations, between the drivers that influence the regrowth of secondary forests (Fig. 1; Supplementary Note 5 and Supplementary Fig. 13). For example, in the South-East and North-East regions, regrowth rates are approximately 50% lower compared to both the two western regions,

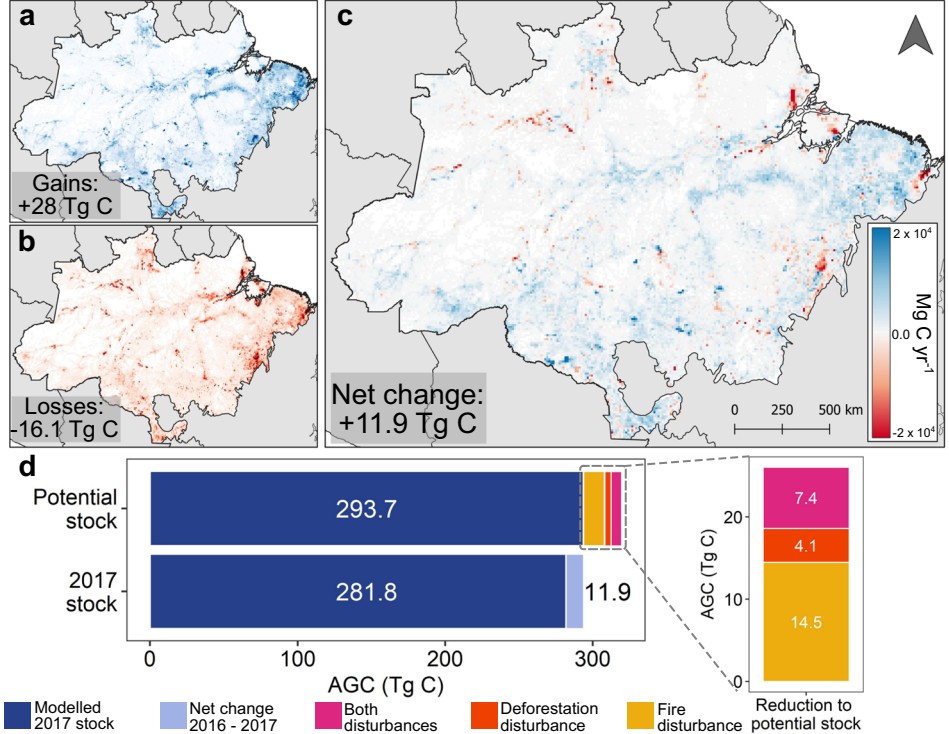

**Fig. 4 The net change in secondary forest carbon stock between 2016 and 2017 and the potential 2017 carbon stock without disturbance.** This shows **a** the Aboveground Carbon (AGC) gains from secondary forest growth and recruitment of new secondary forests, and **b** the losses from reduction in secondary forest area due to deforestation, to provide **c** a net change in the carbon stock between 2016 and 2017 per 0.1° grid. In Panel **d** the bottom bar shows the total carbon stock in 2016 (dark blue) with the net gain in carbon stock in 2017 (light blue). Panel **d** top bar shows the total carbon stock in 2017 (dark blue) and the potential carbon stock that would have been accumulated in secondary forests from their initial establishment (post the start of the study period in 1985) up until 2017 assuming none of the forest experienced any kind of disturbance (burning or repeated deforestation cycles prior to regrowth), using the region-specific regrowth models from non-disturbed secondary forests developed in this study.

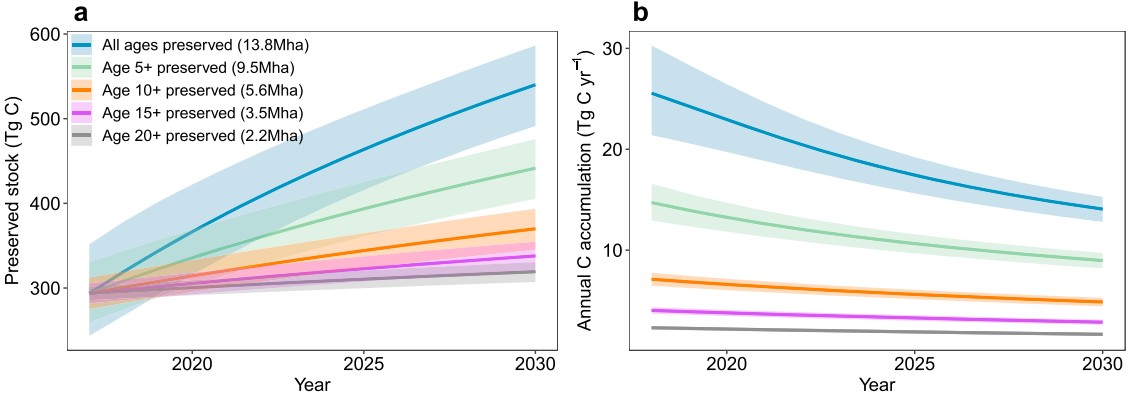

**Fig. 5 The potential carbon stock and annual carbon accumulation in Brazilian Amazonian secondary forests from 2017 to 2030 based on different levels of protection of the 2017 secondary forest area.** The changes to **a** the carbon stock, and **b** the annual carbon accumulation, are estimated for the coming decade, considering different scenarios of preservation of secondary forest area. Calculated using the region-specific regrowth models developed in this study. Shading denotes the 95% confidence interval of the regrowth model.

likely due to the hydro-climatic conditions which reduce growth (low precipitation, ~1913 mm yr$^{-1}$; very high MCWD, ~−325.5 mm yr$^{-1}$; moderate SW radiation, ~181.7 Wm$^{-2}$). Furthermore, this drier environment is more susceptible to burning, which reduces regrowth rates even further (Fig. 3d) and causes loss of carbon stocks through emissions from burning and post-fire mortalities.

Previous field-based studies have estimated the reduction in regrowth due to fire to be 50% (reducing from 3.2 to 1.7 Mg C ha$^{-1}$ yr$^{-1}$)[16], which is similar to the average reduction estimated in our study (40%). With our method we were able to provide additional information disaggregated by regions, showing that the regrowth rates in the North-West and South-West regions, were 20% and 60% lower, respectively, for secondary forests exposed to fire compared to those that were non-disturbed (Supplementary Table 9). We might expect a higher impact of burning in the North-West, where the forest species are not adapted to burning and therefore mortality might be higher and regeneration lower. However, the lower overall impact of fire disturbance on the regrowth rate seen in the North-Western region is likely linked to

the wet climate in this region (Supplementary Fig. 9b), which reduces the intensity and occurrence of the burning itself (Fig. 3a). This would explain why the regrowth rate is hardly affected by fire, and why it is identified as the least important driver in this region (Fig. 2b). The distinct regrowth rates considering the interactions between the drivers highlights the complexity of secondary forest regrowth regimes that cannot be represented by single biome-wide regrowth models. Additionally, the spatial extent of fire disturbance is likely to be more widespread than presented in our study, as the remote sensing product, based on automatic detection, used in this study underestimates burnt area by ~25% compared to manual photointerpretation methods[40].

Previous research has shown that young secondary forests in the central and North-Eastern Amazon have a higher wood density compared to secondary forests in the North-West, which results in a higher overall carbon stock in the long term[34,41] (Supplementary Table 9). Indeed, in our study, the highest median AGC of old-growth forests were in the North-East (135.5 Mg C ha$^{-1}$). While this pattern is similar, the absolute values are up to 30% lower compared to other studies (up to 200 Mg C ha$^{-1}$)[42,43]. Conversely, the lower wood density secondary forest in the North-West have a larger carbon assimilation rate[34], which is reflected in the high regrowth rates calculated in our study for this region. Wood density, and by extension species composition may therefore be a hidden driver of regrowth rates, influenced by the climate variables used in our study. We expect this pattern to become clearer in future studies, which explore secondary forest regrowth rates across the entire Amazon rainforest.

Across Amazonia, we found fire was one of the most important drivers affecting secondary forest regrowth. Other studies have also shown the importance of fire in influencing regrowth[16,44,45], but they have not quantitatively assessed the relative importance by region. Fire is most important in the Southern, drier, regions, including the so-called Arc of deforestation, an area prone to land clearing by fires (Fig. 2d, e). The other anthropogenic driver, repeated deforestations, was evaluated as the least important driver affecting secondary forest regrowth across Amazonia (Figs. 1g; 2b–e). The low importance estimated in our study may be a local-scale artefact which our model cannot account for in the large climate regions identified in this study (Fig. 2a). Environmental drivers act on regional scales and influence forest type and species physiology. Both fire and deforestation typically act on the local scale. In recent decades, the scale at which deforestation events occur has decreased even further, with more very small-scale (<1 ha) deforestation events being observed[46]. If the scale of deforestation is smaller than the resolution of our deforestation detection method, the event may not be accounted for, but will still be reflected in the AGC signature.

We observe the combined effect of fire and repeated deforestation to reduce the regrowth rates the most (Fig. 3; Supplementary Table 9). Both disturbances negatively impact secondary forest regrowth by reducing the seed bank, natural biodiversity, soil nutrient and water availability, which can cause arrested succession (a disturbance preventing the natural successional growth)[47]. We see evidence of arrested succession in the slow growth (up to 80% lower) and early plateau in AGC (12–25 years) in some regions that experienced successive disturbance and suboptimal environmental conditions (Fig. 3b, d; Supplementary Table 9).

Given that our study consisted of 32 years of secondary forest data and one year of AGC data, each of which has associated uncertainties, we take caution with the regrowth rates modelled much beyond this period (Supplementary Note 3). However, the results highlight the potential threat that an alternative stable state, of low AGC in older secondary forests, could arise if they are not managed sustainably and experience successive disturbance[48,49]. Even in the regions of no disturbance and favourable environmental conditions, where secondary forest AGC recovered to old-growth forest levels up to 4 times more rapidly, we estimated the minimum time taken to reach old-growth forest AGC to be ~100 years (Supplementary Tables 8 and 9). Secondary forests will therefore not replace old-growth forests on policy-relevant timescales, stressing the continued need to conserve existing old-growth forests (Supplementary Table 9)[50].

Undisturbed, old-growth forests not only serve to maintain the current carbon sink but also act as key sources of seeds for regeneration. However, disturbances to both old-growth and secondary forests have increased the proportion of low wood density and small-seeded tree species[51]. Identifying the proximity of secondary forests to disturbed versus undisturbed forests could potentially be another driving variable impacting the regrowth rates we have calculated in this study. Datasets that differentiate disturbed from non-disturbed forests are only becoming available now[52]. At present it is estimated that just 13% of Amazonian secondary forests are within 1 km proximity to areas with >80% old-growth forest[53], but whether these forests are disturbed remains unclear. Recent research has shown that proximity to young forests also results in faster forest-cover recovery and more species rich regeneration[54]. This would suggest that the overall success of secondary forest regeneration may in part be linked to preserving surrounding (regrowth) forests too. The assessment of secondary forest proximity to both other secondary forests as well as disturbed and undisturbed old-growth forests goes beyond the scope of the current study, but could be applied in future analysis, highlighting the potential of the method used in this study.

Fragmentation of forests increases their vulnerability to fire and other climate extremes such as drought[55] and is exacerbated by additional land-use changes, which increase fragmentation even further[56,57]. This is problematic for two reasons. Firstly, the majority of current secondary forests are fragmented, isolated from other forested landscapes[53]. Secondly, the threat of forest water deficit and, consequently, drought-induced fire disturbances are predicted to increase into the 21st Century due to ongoing climate change[35]. Research has shown that drought increases; stem and seedling mortality, reducing regrowth and regeneration, respectively[56]. This threat is highest during early succession, when the low, open canopy of the forest area makes them susceptible to higher temperatures and drying[56]. If the predicted 21st Century climate-scenario arises, the reduced regrowth rate of secondary forests as seen in the already dry (1913 mm yr$^{-1}$ precipitation) and water deficient (−328.5 mm yr$^{-1}$ MCWD) South-East region in our analysis is likely to be more widespread and severe (Figs. 1–3). Moreover, there has been a slow shift to more dry-affiliated Amazonian tree genera[58], which have a lower biomass and are more savannah-like in nature[59] as some species reach their adaptive limits to ongoing drier conditions[60]. Such a shift would threaten the permanence of the carbon sequestration potential of secondary forests as we have calculated in this study, especially if changes in tree communities lag substantially behind climatic changes[17,58].

Given that some degree of 21st century climate change is now already inevitable, it is imperative to limit anthropogenic disturbances, such as fire and deforestation in order to avoid further reductions in forest regrowth and forest carbon stocks. Overall, we estimate these disturbances to have contributed to an 8% reduction in the total potential 2017 carbon stock since 1985 (Fig. 4d), with the highest relative reduction (11%) in North-Eastern Amazonia (Supplementary Fig. 12). This has implications for policies concerning human-induced burning regimes and deforestation of secondary forests. Our analysis has shown that

avoiding these actions increases the regrowth potential of secondary forests.

Secondary forest regrowth can help Brazil to achieve its NDC goals of reducing net national emissions by 37% in 2025 and 43% in 2030 compared to 2005 levels (2.1 GtCO$_2$e yr$^{-1}$)[7]. These targets are equivalent to net emissions of 361 Tg C yr$^{-1}$ (1.3 GtCO$_2$e yr$^{-1}$) and 326 Tg C yr$^{-1}$ (1.2 GtCO$_2$e yr$^{-1}$) in the year 2025 and 2030, respectively. We model the future carbon sequestration rate by preserving all standing secondary forests and find that the annual carbon accumulation would be equivalent to providing an additional ~5.5 ± 0.7% reduction to the 2025/2030 emissions target (Fig. 5b). Conversely, if only secondary forests older than 20 years in 2017 were preserved, the additional mitigation potential would reduce to less than 1% (Fig. 5b). The modelling shows that various levels of secondary forest preservation can contribute significantly to Brazil reaching its NDC targets, regardless of any changes made to future NDC revisions. However, these estimates assume that future rates of deforestation in secondary and old-growth forests remain sustainable.

In recent years, emissions from deforestation in Amazonia have accelerated to levels approximately equal to the beginning of the 21st century (170 Tg C yr$^{-1}$ in 2019[15,61]). Assuming all secondary forests standing in 2017 still stood in 2019, the annual secondary forest carbon accumulation would have offset 14 ± 1% of the gross carbon emissions from Amazonian deforestation in that year. The climate mitigation potential of secondary forests within the Brazilian NDC for the next decade can therefore only be realised if a sustainable management of all forests is achieved now. This highlights the carbon benefits of urgent actions to implement legal mechanisms that protect secondary forests on a national scale[20], which would bring with it multiple co-benefits such as enabling biodiversity recovery[62].

The models developed in our study provide a new assessment of the carbon sink potential of secondary forests in the Brazilian Amazon, considering age, persistence, local and regional drivers. This type of approach using regional and global remote sensing products has not been attempted before to such a high spatial resolution. The models have the potential to benefit both the carbon modelling and carbon-policy communities to help understand the regional variations of regrowth under different drivers. The carbon modelling community will benefit from the ability to spatially monitor carbon dynamics, which can be incorporated into models and scenarios of land cover and climate change. Our models provide a new, detailed quantification of the naturally regrowing secondary forest carbon sinks. This will benefit carbon-policy communities by helping to assess locations for restoring and reforesting 12 Mha of forests, as proposed by Brazil's NDC, that would maximise regrowth and thereby be most beneficial to mitigating climate change. This includes areas with limited anthropogenic disturbances, which will minimise forest restoration and thereby costs of implementation and conservation. However, the success of naturally regrowing secondary forests as means for restoration can only be realised by legally designating land for restoration, monitoring and protection. Additionally, the results can be used to improve monitoring under the Reducing Emissions from Deforestation and Degradation (REDD+) scheme. This approach would not be limited to Amazonia and could be applied in other countries where field data may be limited.

The drivers used in this study to assess regrowth potential can be developed further to include other important variables that influence regeneration and regrowth. This includes variables such as the proximity to other forest landscapes, both young and old and disturbed and undisturbed[51,54] as well as the type of previous land-use practices (livestock, agriculture and forestry) and the period of active land use before abandonment[53]. For instance,

secondary forests regrow 38% faster on land used for agriculture than those for cattle pastures[44,63]. Our method has provided an initial steppingstone to assessing the drivers impacting secondary forest regrowth in a spatial manner and shown the potential of utilising a combination of remote sensing products in a space-for-time substitution approach.

Our study has quantified the varied and complicated regrowth rates of secondary forests by multiple drivers across Amazonia. Given the uncertain and potentially threatened status of old-growth forest sinks due to ongoing climate change[4], it is imperative to limit human-induced fire and deforestation disturbance in both old-growth and secondary forests. By preserving the remaining old-growth forest stock and sustainably managing secondary forests we can maintain and increase the carbon sink of this globally important biome and help it to achieve its climate mitigation potential.

## Methods

**Identifying areas of secondary forest and their ages**. The underlying product for this research was the land-use and land-cover product (MapBiomas Collection 3.1), available for the whole of Brazil for the years 1985–2017[25]. The dataset is based on Landsat image classification, mapping annual land use and land cover at 30 m spatial resolution. We follow a very similar methodology applied by recent studies[10,18,53,64] to identify areas of secondary forests and determine their respective ages. We reclassified forest land and all land under human use to values of 1 and 0, respectively, and tracked, when a conversion from anthropogenic (0) to forest land (1) took place. Consecutive years following this transition in which a forest remained forest, were classed as secondary forest and used to estimate their respective ages (in years). Ages ranged from 1 to 32 years since the MapBiomas product (v3.1) is available for 1985–2017. Any forest land pixels that did not undergo a transition during this period were considered an old-growth forest. A limitation is therefore that this method cannot classify forests as secondary forest that were deforested and regrew before 1985. If an area of secondary forest was deforested during the period of analysis, we disregarded the secondary forest area and only began calculating the age again if a conversion from 0 to 1 took place. From this we also calculated the number of times an area of secondary forest was deforested prior to the most recent regrowth (termed 'repeated deforestations' throughout the text) during the period 1986–2016.

Previous research has shown that the MapBiomas product misclassifies perennial crops such as oil palm plantations[10] and other plantation forests as natural forests (Supplementary Fig. 2). To remove misclassified areas, we used the latest land-cover data of another, widely used Brazilian land-cover product, TerraClass-2014[9]. Finally, we excluded areas of secondary forest (within a 3 km radius) that overlay field inventory sites of secondary forest for cross validation of our method (Supplementary Fig. 10; Supplementary Table 10).

**Modelling carbon sequestration with different drivers**. To model the regrowth of secondary forests we applied a space-for-time substitution method. Instead of tracking the associated Aboveground Carbon (AGC) regrowth over time, the regrowth was estimated by considering the available ages of the standing secondary forest area in 2017 and the associated AGC at the same time. Here we explain the methods used to determine secondary forest AGC using the ESA-CCI Aboveground Biomass (AGB) product (100-m) for the year 2017[23] (see Supplementary Notes 1 and 2). All analysis was carried out in the original product units (AGB) but expressed as AGC by assuming a 2:1 ratio of biomass to carbon[24].

The ESA-CCI AGB product was only released in late 2019 and was in its early phases of development at the time of use. However, given that its spatial resolution was high enough to separate areas of only secondary forest and its recent acquisition warranted its use for this research. Only areas of secondary forest greater than 9000 m$^2$ were considered for further analysis, an area approximately equal to 1 pixel of the ESA-CCI product. Despite limiting the study to these larger secondary forest polygons, we were still left with just under 2.5 million polygons of secondary forest to analyse. The secondary forest map was laid over the AGC data and the modal AGC was extracted for each secondary forest polygon using the "zonal_stats" function available in the "rasterstats" module for the programming language "Python" (v3.6). We then aggregated the AGC values by the age of secondary forest and used the median AGC value for each age in further analysis. We applied a bias correction to the median AGC values, subtracting the smallest median value from all values to shift the data to begin at or near 0 Mg C ha$^{-1}$ AGC for a 1-year-old secondary forest.

Following this, we used six remote sensing products of driving variables widely accepted to influence regrowth of forests. The data products included four environmental drivers (1–4) and two anthropogenic disturbance drivers (5–6): (1) Mean annual downward shortwave radiation (for the period 1985–2017)[26], (2) Mean annual precipitation (for the period 1985–2017)[27], (3) the mean Maximum Cumulative Water Deficit (MCWD) (for the period 1985–2017)[65,66], (4) Soil

Cation Concentration[30], (5) Annual burned areas (between 2001 and 2017)[31] and (6) Number of times a secondary forest area was deforested between 1987 and 2017 (repeated deforestations) (this study). These products all have different spatial resolutions (Supplementary Table 1) and so had to be resampled to the size of secondary forest pixels (30-m spatial resolution) using the "resample" package in the Geographic Information System programme, ArcMap10.6. We calculated the key zonal statistics of these variables such as the mean value of the driver affecting a specific area of secondary forest, again using the "zonal_stats" function in Python.

The drivers were then grouped according to numerical limits, such as the 25, 50 and 75th percentiles. We then modelled the AGC for the age of secondary forest under these groupings using the commonly used Chapman-Richard model for regrowth[67]:

$$Y_t = A\left(1 - e^{-kt}\right)^c \pm \varepsilon; A, \, k \, \text{and} \, c > 0 \qquad (1)$$

where $Y_t$ refers to the AGC at age $t$; A is the AGC asymptote or the AGC of the old-growth forest; $k$ is a growth rate coefficient of $Y$ as a function of age; $c$ is a coefficient that determines the shape of the growth curve; and $\varepsilon$ is an error term. We applied the "nls" function available in the "nlstools" package for the statistical software R (v4.0.2)[68,69]. We assumed that after a given amount of time, the AGC could return to levels equivalent to old-growth forests, and reach a pre-calculated asymptote. As such, we extracted the median, bias-corrected AGC value of old-growth forests under each variable condition from the ESA-CCI AGC product to represent the value of the asymptote (Supplementary Fig. 6 and Supplementary Tables 8 and 9). From this, we could also determine if and when the modelled AGC of secondary forest regrowth would reach those equivalent to old-growth forest levels. Forcing the models to "fit" to an expected value for the asymptote value naturally increases the error of our model, partly due to heterogeneity in old-growth forest values within each variable condition.

**Determining the importance of each driver.** We used an ensemble machine learning algorithm, the so-called "random forest" model to assess which of the drivers used in this research were the most important in influencing the regrowth of secondary forests. We carried out all analysis using the conditional random-forest model "cforest" available in the predictive model package "caret" in R. The "cforest" random-forest model provides more accurate importance estimates compared to more traditional random-forest models such as "randomForest" when the dataset includes both continuous (e.g. precipitation) and categorical data (e.g. burnt, not burnt) data[70].

Additionally, the variable importance assessed using the conditional random-forest model better reflects the true impact of each variable, regardless of any correlation between the variables, compared to more traditional random-forest models[71,72]. In any geospatial analysis such as in this study, the variables are likely to be spatially correlated (Supplementary Note 5; Supplementary Fig. 13). It was therefore important to use a model that is not biased towards correlated variables. An important consideration when applying the conditional random-forest analysis on such big data is finding the balance between accuracy and computational efficiency. We follow the approach by Behnamian et al.[73] and apply the random-forest model over a smaller but multiple (30) randomly selected sample sizes, and using a smaller number (500) of decision trees. This approach has been found to result in a stable mean importance analysis, and is more computationally efficient than running a single random-forest model over a larger sample size or using a higher number of decision trees[73].

We applied a stratified random sample equating to 0.1% of the data into the random-forest model ($n = 2500$). This sample size was more than the minimum number of samples needed ($1000 = 0.04\%$) to ensure results would be within the 95% confidence interval with a sampling error of 5% using a multinomial function[74]. We used 80% of the sampled data for training the model and the remaining 20% to test the model. From this analysis we estimated the "conditional permutation importance" for each variable. Following all the iterations, we take the average importance across the model runs and express the importance of each variable as a ranking. This is because the interpretation of the results should be limited to the rankings and not the absolute values of the importance assessment[71].

**Representing spatial patterns of secondary forest regrowth.** We created a regional classification based on the three climate variables driving regrowth (SW radiation, precipitation, and MCWD). We used an unsupervised K-means cluster analysis to group Amazonia into regions based on similarities between the secondary forests in terms of the drivers' variability. We then subclassified each region based on the type of disturbance (fire and/or deforestation) experienced by the secondary forest. The aim of this was to show areas of secondary forest that experience similar conditions and the effect this has on regrowth in a spatially explicit manner. We developed 16 regional models of regrowth and included the median, bias-corrected AGC value for old-growth forest in each of the regions as the asymptote of the models. Using the random-forest model, we again determined the importance of drivers in each region, as described in the previous section, this time using a sample size of 1500 in each region.

**Estimating 2017 carbon stock and future carbon sinks.** We estimated the 2017 carbon stock by applying the corresponding regional models to all pixels initially identified as secondary forest with respect to the pixel age, and whether the pixel experienced any disturbances. From this we were able to estimate the carbon stock in 2017 for all secondary forests and the net carbon change from 2016 to 2017. We also considered an alternative scenario in which no forest disturbance occurred during regrowth by applying the no-disturbance models to the corresponding regions. In this alternative scenario, we were able to calculate the resulting potential 2017 carbon stock and associated reduction due to disturbances. Finally, we applied a similar approach as Chazdon et al.[14] and aged the standing secondary forest in 2017 to model the carbon stock and annual carbon accumulation for the next decade considering different scenarios of secondary forest preservation: (1) all forests; (2) forest with ages 5+; (3) forest with ages 10+; (4) forest with ages 15+; (5) forest with ages 20+ years.

## Data availability

The original data used in this study are all publicly available from their sources: MapbiomasV3.1 (https://mapbiomas.org/); ESA-CCI Aboveground Biomass for the year 2017 (https://catalogue.ceda.ac.uk/uuid/bedc59f37c9545c981a839eb552e4084); CHIRPS precipitation data: Funk et al.[27]—https://edcintl.cr.usgs.gov/downloads/sciweb1/shared/fews/web/global/monthly/chirps/final/downloads/monthly/); Soil Cation Concentration data: Zuquim et al.[30]—https://doi.pangaea.de/10.1594/PANGAEA.879542; Shortwave radiation data: TerraClimate, Didan[31], http://www.climatologylab.org/terraclimate.html. The MCWD data can be produced using data from Funk et al. combining with code available from Campanharo and Silva Junior[66]—https://zenodo.org/record/2652658#.X9CV-aFxdPY. The processed data and products produced in this research are available on the following repository: https://zenodo.org/record/4479234#.YBVdBHNxdPY[75]. The regrowth models can be built by users using Equation 1 in the methods section and information provided in Supplementary Tables 8 and 9 and the shapefiles corresponding to the four climate regions are available in the aforementioned repository. Additional map data, namely the Amazon biome region and country vector shapefiles are available from the following two sources, respectively: http://terrabrasilis.dpi.inpe.br/en/home-page/ and https://www.naturalearthdata.com/downloads/10m-physical-vectors/10m-land/. Finally, the TerraClass land-cover dataset is available from: https://www.terraclass.gov.br/. Source data are provided with this paper.

## Code availability

The code used to extract secondary forest age and extent is available from Silva Junior et al. (2020)[18]. The code used to calculate the MCWD index is available here: https://zenodo.org/record/2625903#.X701CulxdPY. The code that was developed in this study is available on GitHub and has a DOI repository with Zenodo: https://github.com/heinrichTrees/secondary-forest-regrowth-amazon-public (https://doi.org/10.5281/zenodo.4479398)[76].

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

## Acknowledgements

V.H.A.H. was supported by a NERC GW4+ Doctoral Training Partnership studentship from the Natural Environment Research Council (NE/L002434/1). R.D. was supported by CNPq (National Council for Scientific and Technological Development) Grant no. 160286/2019-0, and FAPESP (São Paulo Research Foundation) Grant no. 2019/21662-8. H.L.G.C. was supported by FAPESP (2018/14423-4). C.T.A. was partially supported by CNPq (140502/2016-5). S.S. and T.M.R. were partly supported by the European Space Agency Climate Change Initiative ESA-CCI RECCAP2 project (ESRIN/ 4000123002/18/I-NB), and the Newton Fund through the Met Office Climate Science for Service Partnership Brazil (CSSP Brazil). The latter provided V.H.A.H. travel support during this study. C.T.A. was partially supported by CNPq (140502/2016-5). C.H.L.S.J. was supported in part by CAPES (Coordenação de Aperfeiçoamento de Pessoal de Nível Superior – Brazil)—Finance Code 001. W.C. was supported in part by CAPES—Finance Code 001 as well as by CNPq (140261/2018-4). J.I.H. was supported in part by funding from NERC (NE/P019765/1). M.A. was supported by a Royal Society Newton Advanced Fellowship (NAF/R1/180405). L.O.A. was supported by the Inter-American Institute for Global Change Research (IAI) (process: SGP-HW 016), FAPESP (process 2016/02018-2, 19/05440-5) and CNPq (442650/2018-3, 441949/2018-5). L.E.O.C.A. was supported by CNPq (processes 305054/2016-3 and 442371/2019-5) and FAPESP (process 2018/15001-6; ARBOLES Project). We are thankful to the developers of all the products used in this study for providing free, open-access datasets.

## Author contributions

V.H.A.H. and L.E.O.C.A. developed the concept and main methodological process. V.H.A.H. carried out the data analysis and wrote the initial manuscript draft with additional support from R.D., H.L.G.C. and T.M.R. H.L.G.C. compiled the field inventory data and provided methodological suggestions. R.D., H.L.G.C., T.M.R., C.T., C.S. and W.C. provided codes that were adapted for this research. J.H., S.S., T.H., T.M.R., M.A. and L.O.A. provided vital comments on the data analysis and scientific support throughout the study. All authors discussed results and provided comments during the preparation of the manuscript.

## Competing interests

The authors declare no competing interests.
