## [Peer Review File · Nature Communications]

REVIEWER COMMENTS

Reviewer #2 (Remarks to the Author):

The manuscript entitled "Large carbon sink potential of Amazonian Secondary Forests to mitigate climate change" present a relevant study that aimed to quantify the contribution of the Secondary Forest to carbon sequestration and the influence of environmental and disturbance drivers on the rate and spatial patterns of regrowth in the Brazilian Amazon. The study addresses a very important topic for spatially monitor carbon dynamics in the tropical forest, and with the method used they provided information disaggregated by regions, with applicability in different sectors of society, such as science, economics and sustainable development.

The paper is very well written and uses current literature. As I am not a native English speaker, I cannot evaluate English.

The statistical approach seems adequate and sound and the methods were described in a clear and detailed way.

The article will be of great interest to the audience of this magazine, but I think it needs to point out more explicitly what are the novel in it, since part of the findings shown in it have already been presented by Poorter et al., 2016 (doi: 10.1038 / nature16512) and Patton et al. 2020 (doi.org/10.1038/s41586-020-2686-x).

I suggest incorporating in the discussion information about how frationation can affect regrowth in FS and whether the change in species composition, already mentioned in the literature, driven by climate change will affect this regrowth.

The manuscript represents a solid piece of work that should be published after some minor corrections.

Reviewer #3 (Remarks to the Author):

I really enjoyed reading this manuscript, which takes the next step to project the impacts of secondary forest regrowth and subsequent disturbances on rates of carbon storage in the Brazilian Amazon. The key advance made in this study was examining the spatial variation in trajectories and drivers of biomass accumulation across this enormous and geographically complex region. The analyses revealed strong differences in these trajectories and also strong effects of factors that have not previously been modelled, such as short-wave radiation and fire disturbances. This study provides a state-of-the-art assessment of the carbon sink potential of regrowth forests, taking into account forest age, persistence, and local drivers of forest regrowth. It will have a major impact on how we assess this important nature-based solution for climate change mitigation.

Specific comments:

Title: I suggest referring to Brazilian Amazon in the title. The study does not cover the entire Amazon basin.

Line 33: this new paper provides global estimates, but doesn't have good spatial resolution within Amazonia

Cook-Patton, S. C., S. Leavitt, D. Gibbs, N. L. Harris, K. Lister, K. J. Anderson-Teixeira, R. D. Briggs, R. L. Chazdon, T. W. Crowther, P. W. Ellis, H. P. Griscom, V. Herrmann, K. D. H. Holl, R. A. Houghton, C. Larrosa, G. Lomax, R. Lucas, P. Madsen, Y. Malhi, A. Paquette, J. D. Parker, K. Paul, D. Routh, S. Roxburgh, S. Saatchi, J. van de Hoogen, W. S. Walker, C. E. Wheeler, S. Wood, L. Xu, and B. W. Griscom. 2020. Mapping carbon accumulation potential from global natural forest regrowth. *Nature* 585: 545-550.

Line 77: There are also key differences in species composition and wood density between western and eastern Amazonia that influence biomass and carbon storage potential. Species composition could be a "hidden" variable that is also correlated with climate variables and affects carbon

storage during forest regrowth:

Baker, T. R., O. L. Phillips, Y. Malhi, S. Almeida, L. Arroyo, A. Di Fiore, T. Erwin, T. J. Killeen, S. G. Laurance, W. F. Laurance, S. L. Lewis, J. Lloyd, A. Monteagudo, D. A. Neill, S. Patino, N. C. A. Pitman, J. N. M. Silva, and R. V. Martinez. 2004. Variation in wood density determines spatial patterns in Amazonian forest biomass. *Global Change Biology* 10:545-562.

Line 90-95: Are these variables spatially correlated? I would expect high SW radiation to be associated with low precipitation and higher CWD. The rank order for these variables is the same for Figure 1 a, b, and c.

Line 111-112: this only takes biophysical conditions into account. What about factors that affect regeneration potential itself, rather than aboveground carbon density (proximity to forests, slope?). Different factors may predict presence/absence of regenerating forest as compared to qualities of regenerating forests.

Line 151-153: Presumably, OG forest biomass also differ regionally. Papers by Phillips and colleagues suggests this is the case. Would be informative here.

Nogueira, E. M., P. M. Fearnside, B. W. Nelson, R. I. Barbosa, and E. W. H. Keizer. 2008. Estimates of forest biomass in the Brazilian Amazon: New allometric equations and adjustments to biomass from wood-volume inventories. *Forest Ecology and Management* 256:1853-1867.

Line 166: not to mention direct carbon emissions from the fire itself

Line 173-174: be specific here: drivers of what? carbon sink in young, regrowth forest?

Line 201-202: A few studies have been published that focus on effects of climate change on tropical forest regrowth and should be mentioned:

Uriarte, M., N. Schwartz, J. S. Powers, E. Marín-Spiotta, W. Liao, and L. K. Werden. 2016. Impacts of climate variability on tree demography in second growth tropical forests: the importance of regional context for predicting successional trajectories. *Biotropica* 48:780-797.

Uriarte, M., J. R. Lasky, V. K. Boukili, and R. L. Chazdon. 2016. A trait-mediated, neighbourhood approach to quantify climate impacts on successional dynamics of tropical rainforests. *Functional Ecology* 30:157-167.

Line 251-253: make sure this message is in the summary and press release!

Line 262: also see this study using MapBiomass data for mapping natural regeneration in Atlantic Forest in Brazil:

Crouzeilles, R., H. L. Beyer, L. M. Monteiro, R. Feltran-Barbieri, A. C. Pessôa, F. S. Barros, D. B. Lindenmayer, E. D. Lino, C. E. Grelle, and R. L. Chazdon. 2020. Achieving cost-effective landscape-scale forest restoration through targeted natural regeneration. *Conservation Letters*:e12709.

Line 269: could data from previous studies be used to estimate this?

Fearnside, P. M., and W. M. Guimaraes. 1996. Carbon uptake by secondary forests in Brazilian Amazonia. *Forest Ecology and Management* 80:35-46.

Helmer, E. H., M. A. Lefsky, and D. A. Roberts. 2009. Biomass accumulation rates of Amazonian secondary forest and biomass of old-growth forests from Landsat time series and the Geoscience Laser Altimeter System. *Journal of Applied Remote Sensing* 3:1-31.

Line 295: did you examine the relationships among these variables (spatial co-variation)?

Line 316: How do OG forests vary geographically in their aboveground carbon density?

Line 352-355: This is similar to the approach used by Chazdon et al. (2016) (reference 14)

Signed,
Robin Chazdon

Responses to Reviewers of the manuscript entitled:

“Large carbon sink potential of Secondary Forests in Brazilian Amazon to mitigate climate change”

By Heinrich V.H.A. and co-authors

First, I hope the Reviewers and their families are well and healthy. We thank both Reviewers for their comprehensive review and supportive comments. We have addressed them in full, which has helped to improve our study. Below, we repeat all Reviewers' comments and reply to the comments one by one. Each comment is numbered, with our responses in bold. The Reviewers were numbered 2 and 3, so we have kept these names to limit confusion, note – there was no response from Reviewer 1 with any changes for us to make.

Kind regards,

Viola Heinrich and co-authors (17/12/2020)

Reviewer #2

1 - The manuscript entitled "Large carbon sink potential of Amazonian Secondary Forests to mitigate climate change" present a relevant study that aimed to quantify the contribution of the Secondary Forest to carbon sequestration and the influence of environmental and disturbance drivers on the rate and spatial patterns of regrowth in the Brazilian Amazon. The study addresses a very important topic for spatially monitor carbon dynamics in the tropical forest, and with the method used they provided information disaggregated by regions, with applicability in different sectors of society, such as science, economics and sustainable development.

The paper is very well written and uses current literature. As I am not a native English speaker, I cannot evaluate English.

The statistical approach seems adequate and sound and the methods were described in a clear and detailed way.

Response: We thank Reviewer #2 for their positive assessment of the paper. We are glad the inter-disciplinary applicability of the paper comes across and that the methods were described and carried out in an adequate manner.

2 - The article will be of great interest to the audience of this magazine, but I think it needs to point out more explicitly what are the novel in it, since part of the findings shown in it have already been presented by Poorter et al., 2016 (doi: 10.1038 / nature16512) and Patton et al. 2020 (doi.org/10.1038/s41586-020-2686-x).

Response: We agree that the novel nature of the paper can be made even more explicit. We have changed the last paragraph in the introduction to highlight the new aspects

our work addresses relative to previous work, including very recent literature such as Patton et al., 2020, the paragraph reads:

“The primary aim of this study is to provide key advances in understanding the spatial variation of secondary forest regrowth in the Brazilian Amazon, a large and geographically complex region. Previous studies have already provided the first steps to understanding regrowth on a biome scale, influenced by some driving variables^{11,14,22}. Here we introduce additional environmental and anthropogenic disturbance drivers that affect regrowth, including local-scale drivers, and for the first time, disaggregate their effects using a spatially explicit approach^{11,14,22}.”

Throughout the results, discussion, and conclusion we have also imbedded further sentences highlighting the novel nature of the work. For example, the beginning of the conclusion now reads:

“The models developed in our study provide a new assessment of the carbon sink potential of secondary forests in the Brazilian Amazon, considering age, persistence, local and regional drivers. This type of approach using regional and global remote sensing products has not been attempted before to such a high spatial resolution. The models have the potential to benefit both the carbon modelling and carbon-policy communities to help understand the regional variations of regrowth under different drivers. The carbon modelling community will benefit from the ability to spatially monitor carbon dynamics, which can be incorporated into models and scenarios of land cover and climate change. Our models provide a new, detailed quantification of the naturally regrowing secondary forest carbon sinks.”

3 - I suggest incorporating in the discussion information about how fractionation can affect regrowth in FS and whether the change in species composition, already mentioned in the literature, driven by climate change will affect this regrowth.

The manuscript represents a solid piece of work that should be published after some minor corrections.

Response: We assumed the term fractionation was a typo and was referring to “fragmentation” of old-growth forest as well as secondary forest. If this was not the case, we apologize and would ask for additional clarification. We have linked this with responses 4 and 6 from reviewer #3, which we believe was related to a similar issue of proximity to old-growth forests and species composition, respectively.

We hope that the following sentences in the text addresses these comments:

Responding to species composition and climate:

- 1) *“We might expect a higher impact of burning in the North-West, where the forest species are not adapted to burning and therefore mortality might be higher and regeneration lower. However, the lower overall impact of fire disturbance on the regrowth rate seen in the North-Western region is likely linked to the wet climate in this region (Supplementary Fig. 9b), which reduces the intensity and occurrence of the burning itself (Figure 3a). This would explain why the regrowth rate is hardly affected by fire, and why it is identified as the least important driver in this region (Figure 2b). The distinct regrowth rates considering the interactions between the drivers highlights the complexity of secondary forest regrowth*

regimes that cannot be represented by single biome-wide regrowth models. Additionally, the spatial extent of fire disturbance is likely to be more widespread than presented in our study, as the remote sensing product, based on automatic detection, used in this study underestimates burnt area by ~25% compared to manual photointerpretation methods⁴⁰.

Previous research has shown that young secondary forests in the central and North-Eastern Amazon have a higher wood density compared to secondary forests in the North-West, which results in a higher overall carbon stock in the long term^{34,41} (Supplementary Table 9). Indeed, in our study, the highest median AGC of old-growth forests were in the North-East (135.5 MgC ha⁻¹). While this pattern is similar, the absolute values are up to 30% lower compared to other studies (up to 200 MgC ha⁻¹)^{42,43}.^[#3-R7] Conversely, the lower wood density secondary forest in the North-West have a larger carbon assimilation rate³⁴, which is reflected in the high regrowth rates calculated in our study for this region. Wood density, and by extension species composition may therefore be a hidden driver of regrowth rates, influenced by the climate variables used in our study. We expect this pattern to become clearer in future studies which explore secondary forest regrowth rates across the entire Amazon rainforest.”

- 2) *“Research has shown that drought increases stem and seedling mortality, reducing regrowth and regeneration respectively⁵⁶. This threat is highest during early succession, when the low, open canopy of the forest area makes them susceptible to higher temperatures and drying⁵⁶.^[#2-R3-R10] If the predicted 21st Century climate-scenario arises, the reduced regrowth rate of secondary forests as seen in the already dry (1913mm yr⁻¹ precipitation) and water deficient (-328.5 mm yr⁻¹ MCWD) South-East region in our analysis is likely to be more widespread and severe (Figures 1-3). Moreover, there has been a slow shift to more dry-affiliated Amazonian tree genera⁵⁸, which have a lower biomass and are more savannah-like in nature⁵⁹ as some species reach their adaptive limits to ongoing drier conditions⁶⁰. Such a shift would threaten the permanence of the carbon sequestration potential of secondary forests as we have calculated in this study, especially if changes in tree communities lag substantially behind climatic changes^{17,58}.”*

Responding to fragmentation impact on regrowth:

“Undisturbed, old-growth forests not only serve to maintain the current carbon sink but also act as key sources of seeds for regeneration. However, disturbances to both old-growth and secondary forests have increased the proportion of low wood density and small-seeded tree species⁵¹. Identifying the proximity of secondary forests to disturbed versus undisturbed forests could potentially be another driving variable impacting the regrowth rates we have calculated in this study. Datasets that differentiate disturbed from non-disturbed forests are only becoming available now⁵². At present it is estimated that just 13% of Amazonian secondary forests are within 1km proximity to areas with >80% old-growth forest⁵³, but whether these forests are disturbed remains unclear. Recent research has shown that proximity to young forests also results in faster forest-cover recovery and more species rich regeneration⁵⁴. This would suggest that the overall success of secondary forest regeneration may in part be linked to preserving surrounding (regrowth) forests too. The assessment of secondary forest proximity to both other secondary forests as well as disturbed and undisturbed old-growth forests goes beyond the scope of the current study, but could be applied in future analysis, highlighting the potential of the method used in this study.”

Reviewer #3

1 - I really enjoyed reading this manuscript, which takes the next step to project the impacts of secondary forest regrowth and subsequent disturbances on rates of carbon storage in the Brazilian Amazon. The key advance made in this study was examining the spatial variation in trajectories and drivers of biomass accumulation across this enormous and geographically complex region. The analyses revealed strong differences in these trajectories and also strong effects of factors that have not previously been modelled, such as short-wave radiation and fire disturbances. This study provides a state-of-the-art assessment of the carbon sink potential of regrowth forests, taking into account forest age, persistence, and local drivers of forest regrowth. It will have a major impact on how we assess this important nature-based solution for climate change mitigation.

Response: We thank the Reviewer for their very positive overall assessment of the paper and their strong words of support highlighting the importance of the research. We were delighted and encouraged to receive the comments and have responded to the specific comments below.

Specific comments:

2 - Title: I suggest referring to Brazilian Amazon in the title. The study does not cover the entire Amazon basin.

Response: Thank you for this suggestion. We have amended the title to read:

“Large carbon sink potential of Secondary Forests in the Brazilian Amazon to mitigate climate change”

It is still within the word limit of Nature Communications for titles (15/15)

3 - Line 33: this new paper provides global estimates, but doesn't have good spatial resolution within Amazonia

Cook-Patton, S. C., S. Leavitt, D. Gibbs, N. L. Harris, K. Lister, K. J. Anderson-Teixeira, R. D. Briggs, R. L. Chazdon, T. W. Crowther, P. W. Ellis, H. P. Griscom, V. Herrmann, K. D. H. Holl, R. A. Houghton, C. Larrosa, G. Lomax, R. Lucas, P. Madsen, Y. Malhi, A. Paquette, J. D. Parker, K. Paul, D. Routh, S. Roxburgh, S. Saatchi, J. van de Hoogen, W. S. Walker, C. E. Wheeler, S. Wood, L. Xu, and B. W. Griscom. 2020. Mapping carbon accumulation potential from global natural forest regrowth. *Nature* 585: 545-550.

Response: Thank you. We have now cited it in the introduction to strengthen our argument that our study addresses the impact of local factors on regrowth, one of the issues mentioned in the conclusion of the Cook-Patton et al. study that was still outstanding following their study:

“The primary aim of this study is to provide key advances in understanding the spatial variation of secondary forest regrowth in the Brazilian Amazon, a large and geographically complex region. Previous studies have already provided the first steps to understanding regrowth on a biome scale, influenced by some driving variables^{11,14,22}. Here we introduce additional environmental and

anthropogenic disturbance drivers that affect regrowth, including local-scale drivers, and for the first time, disaggregate their effects using a spatially explicit approach^{11,14,22}.

4 - Line 77: There are also key differences in species composition and wood density between western and eastern Amazonia that influence biomass and carbon storage potential. Species composition could be a "hidden" variable that is also correlated with climate variables and affects carbon storage during forest regrowth:

Baker, T. R., O. L. Phillips, Y. Malhi, S. Almeida, L. Arroyo, A. Di Fiore, T. Erwin, T. J. Killeen, S. G. Laurance, W. F. Laurance, S. L. Lewis, J. Lloyd, A. Monteagudo, D. A. Neill, S. Patino, N. C. A. Pitman, J. N. M. Silva, and R. V. Martinez. 2004. Variation in wood density determines spatial patterns in Amazonian forest biomass. *Global Change Biology* 10:545-562.

Response: Again, thank you pointing us to this additional reference and point to consider. We have included an additional paragraph in the discussion as well as embedded sentences throughout that addresses this point (see below). However, we expect to see the impact of species competition and wood density to emerge more over Pan-Amazonia, i.e. beyond the Brazilian Amazon, as this accounts for the larger-leaved, lower wood density trees in the Peruvian, Colombian, Ecuador Amazon (i.e. further West towards the Andes). Expanding beyond the Brazilian Amazon is beyond the scope of the current study, but we take on board this valuable comment for future studies using this approach for the whole Amazon region, and present the amended paragraph in the discussion:

“We might expect a higher impact of burning in the North-West, where the forest species are not adapted to burning and therefore mortality might be higher and regeneration lower. However, the lower overall impact of fire disturbance on the regrowth rate seen in the North-Western region is likely linked to the wet climate in this region (Supplementary Fig. 9b), which reduces the intensity and occurrence of the burning itself (Figure 3a). This would explain why the regrowth rate is hardly affected by fire, and why it is identified as the least important driver in this region (Figure 2b). The distinct regrowth rates considering the interactions between the drivers highlights the complexity of secondary forest regrowth regimes that cannot be represented by single biome-wide regrowth models. Additionally, the spatial extent of fire disturbance is likely to be more widespread than presented in our study, as the remote sensing product, based on automatic detection, used in this study underestimates burnt area by ~25% compared to manual photointerpretation methods⁴⁰.

Previous research has shown that young secondary forests in the central and North-Eastern Amazon have a higher wood density compared to secondary forests in the North-West, which results in a higher overall carbon stock in the long term^{34,41} (Supplementary Table 9). Indeed, in our study, the highest median AGC of old-growth forests were in the North-East (135.5 MgC ha⁻¹). While this pattern is similar, the absolute values are up to 30% lower compared to other studies (up to 200 MgC ha⁻¹)^{42,43}. Conversely, the lower wood density secondary forest in the North-West have a larger carbon assimilation rate³⁴, which is reflected in the high regrowth rates calculated in our study for this region. Wood density, and by extension species composition may therefore be a hidden driver of regrowth rates, influenced by the climate variables used in our study. We expect this pattern to become clearer in future studies which explore secondary forest regrowth rates across the entire Amazon rainforest.”

5 - Line 90-95: Are these variables spatially correlated? I would expect high SW radiation to be associated with low precipitation and higher CWD. The rank order for these variables is the same for Figure 1 a, b, and c.

Response: Thank you for raising this important point, we believe we could have made this analysis clearer. As a result, we carried out some additional analysis assessing the spatial correlation of the driving variables. We used the Spearman's rank to determine the degree of correlation between the drivers that were later used to build the regrowth models. We describe these procedures in the Supplementary Note 5 and present results as Supplementary Figure 13 (see below). This analysis brought to light that some variables are indeed spatially correlated. For example, as expected, precipitation and MCWD are highly (up to $r = 0.8$) correlated given that the CHIRPS precipitation dataset was used to derive the MCWD metric. Other variables showed weak to moderate correlation (up to $r=0.4-0.5$). Given the observed significant correlations, this led us to query whether the random forest analysis, determining the importance of the variables in influencing Aboveground Biomass, would be biased towards spatially correlated ones. Upon further reading (Strobl et al., 2009) we realised that whilst the random forest model used in this study (conditional forest - cforest) is considered more stable than traditional random forest models with regards to addressing variables correlation, this condition needed to be explicitly specified. We therefore re-ran the analysis assessing the importance of the variables in impacting AGC, with the "conditional permutation" parameter set to "True" (see new Supplementary Note 5 and amended Methods section). Overall, the results from the additional analysis showed that Short-wave radiation remained the most important variable, however the importance of fire increases, especially in the Southern Amazon regions. This result is in line with what we expected, especially considering the climate and land-use activities in Southern Amazon, which make this region more fire prone. We are now more confident with the results and the conclusions we draw from this part of the analysis, though the overall conclusions of our study remain the same but have been strengthened by this new analysis. We include the updated Figures 1 and 2 for your reference as well as Supplementary Figure 13 (see below)

Strobl, C., Hothorn, S. and Zeileis, A. (2009) Party on! A New, Conditional Variable Importance Measure for Random Forests Available in the Party Package. Technical Report Number 050, Department of Statistics, University of Munich, Munich.

Figure 1 | Secondary forest carbon accumulation with increasing age under different driving conditions. Drivers are (a) Annual mean downward shortwave radiation (Wm^{-2}), (b) Maximum Cumulative Water Deficit (MCWD; mm yr^{-1}), (c) Annual mean precipitation (mm yr^{-1}), (d) Soil Cation Concentration (SCC; $\text{cmol}(+) \text{kg}^{-1}$), (e) Fire occurrences between 2001 and 2017, and (f) Number of deforestations prior to regrowth between 1985 and 2017, where 1 refers to areas that have only experienced the original conversion from old-growth forest to secondary forests during the period 1985 to 2017 with no subsequent deforestation events. The bar graph (g) shows the average importance ranking of the drivers (a-f), as well as Forest age, in influencing Aboveground carbon (AGC) accumulation. The average ranking was calculated following 30 iterations of a conditional random forest model. The importance has been ranked from least important (1) to most important (7) and the vertical dotted line separates environmental drivers (left) from anthropogenic disturbance drivers (right). Shading in (a-f) denotes the 95% confidence interval of the models, based on the median value of the initial data for each age – dots in figures. The error bars in (g) denote the 95% confidence interval.

Figure 2 | Importance ranking of environmental and disturbance drivers on secondary forest regrowth grouped by climatological regions. (a) Regions are grouped according to similarities in Maximum Cumulative Water Deficit (MCWD), annual average downward shortwave radiation and annual average precipitation. See Supplementary Table 9 for quantitative interpretations of the regions. The average importance ranking for each of the six variables, as well as Forest age, is shown for (b) the North-West region, (c) the North-East and Central-North region, (d) the South-West and Central region, and (e) the South-East and North region. The average ranking was calculated following 30 iterations of a conditional random forest model. The importance has been ranked from least important (1) to most important (7) and the vertical dotted line separates environmental drivers (left) from anthropogenic disturbance drivers (right). The error bars in (b-e) denote the 95% confidence interval.

Supplementary Figure 13 | Spatial co-variation of driving variables across Amazonia. The Spearman's Rank correlation coefficient of different combinations of the driving variables in different regions across Amazonia, where (a) is the North-West region, (b) the North-East and Central-North regions, (c) the South-West and Central region and (d) the South-East and North regions, and (e) the entire biome. Stars denote statistical significance of the coefficient. The number of samples used to assess the coefficient (N) is shown in the figure. Shading denotes the sign and degree of the relationship.

6 - Line 111-112: this only takes biophysical conditions into account. What about factors that affect regeneration potential itself, rather than aboveground carbon density (proximity to forests, slope?). Different factors may predict presence/absence of regenerating forest as compared to qualities of regenerating forests.

Response: We agree. We hope to expand on the approach used in this study to include additional variables, especially proximity to forests, on a broader (pan)-tropical/Amazonian scale in future research. With regards to your point about the impact of slope of the terrain on regeneration, a recent study by Hawes et al. (2020 – see below) showed that slope and (soil clay) had a weak and non-significant effect on seed dispersal and size. While it would be interesting to determine the impact this has on secondary forest regeneration and regrowth we did not consider it here but will consider it in our future analysis that builds on this research.

To address your point in this paper we have included these aspects in the discussion as future directions:

“Undisturbed, old-growth forests not only serve to maintain the current carbon sink but also act as key sources of seeds for regeneration. However, disturbances to both old-growth and secondary forests have increased the proportion of low wood density and small-seeded tree species⁵¹. Identifying the proximity of secondary forests to disturbed versus undisturbed forests could

potentially be another driving variable impacting the regrowth rates we have calculated in this study. Datasets that differentiate disturbed from non-disturbed forests are only becoming available now⁵². At present it is estimated that just 13% of Amazonian secondary forests are within 1km proximity to areas with >80% old-growth forest⁵³, but whether these forests are disturbed remains unclear. Recent research has shown that proximity to young forests also results in faster forest-cover recovery and more species rich regeneration⁵⁴. This would suggest that the overall success of secondary forest regeneration may in part be linked to preserving surrounding (regrowth) forests too. The assessment of secondary forest proximity to both other secondary forests as well as disturbed and undisturbed old-growth forests goes beyond the scope of the current study, but could be applied in future analysis, highlighting the potential of the method used in this study.”

Hawes, J. E. et al. A large-scale assessment of plant dispersal mode and seed traits across human-modified Amazonian forests. *J. Ecol.* 108, 1373–1385 (2020)

7 - Line 151-153: Presumably, OG forest biomass also differ regionally. Papers by Phillips and colleagues suggests this is the case. Would be informative here.

Nogueira, E. M., P. M. Fearnside, B. W. Nelson, R. I. Barbosa, and E. W. H. Keizer. 2008. Estimates of forest biomass in the Brazilian Amazon: New allometric equations and adjustments to biomass from wood-volume inventories. *Forest Ecology and Management* 256:1853-1867.

Response: Indeed it does. Similar to Response 4 we expect this issue to become more clear when we expand the analysis to a Pan-Amazonian one in the future. We have included a paragraph in the Discussion and have referred to the AGC of old-growth forests that were used to help build the models:

“Previous research has shown that young secondary forests in the central and North-Eastern Amazon have a higher wood density compared to secondary forests in the North-West, which results in a higher overall carbon stock in the long term^{34,41} (Supplementary Table 9). Indeed, in our study, the highest median AGC of old-growth forests were in the North-East (135.5 MgC ha⁻¹). While this pattern is similar, the absolute values are up to 30% lower compared to other studies (up to 200 MgC ha⁻¹)^{42,43}.”

8 - Line 166: not to mention direct carbon emissions from the fire itself

Response: This is true. We have added this additional detail. The sentence now reads:

Furthermore, this drier environment is more susceptible to burning, which reduces regrowth rates even further (Figure 3d) and causes loss of carbon stocks through emissions from burning and post-fire mortalities.

9 - Line 173-174: be specific here: drivers of what? carbon sink in young, regrowth forest?

Response: Thank you for noticing this. To improve clarity, we have added the term “in Amazonian secondary forests” to the sentence.

10 - Line 201-202: A few studies have been published that focus on effects of climate change on tropical forest regrowth and should be mentioned:

Uriarte, M., N. Schwartz, J. S. Powers, E. Marín-Spiotta, W. Liao, and L. K. Werden. 2016. Impacts of climate variability on tree demography in second growth tropical forests: the importance of regional context for predicting successional trajectories. *Biotropica* 48:780-797.

Uriarte, M., J. R. Lasky, V. K. Boukili, and R. L. Chazdon. 2016. A trait-mediated, neighbourhood approach to quantify climate impacts on successional dynamics of tropical rainforests. *Functional Ecology* 30:157-167.

Response: We agree that these are very relevant papers, they have been imbedded within the discussion that now includes a detailed paragraph on the impact of Climate Change on species and regrowth:

Fragmentation of forests increases their vulnerability to fire and other climate extremes such as drought⁵⁵ and is exacerbated by additional land-use changes which increase fragmentation even further^{56,57}. This is problematic for two reasons. Firstly, the majority of current secondary forests are fragmented, isolated from other forested landscapes⁵³. Secondly, the threat of forest water deficit and, consequently, drought-induced fire disturbances are predicted to increase into the 21st Century due to ongoing climate change³⁵. Research has shown that drought increases stem and seedling mortality, reducing regrowth and regeneration respectively⁵⁶. This threat is highest during early succession, when the low, open canopy of the forest area makes them susceptible to higher temperatures and drying⁵⁶. If the predicted 21st Century climate-scenario arises, the reduced regrowth rate of secondary forests as seen in the already dry (1913mm yr⁻¹ precipitation) and water deficient (-328.5 mm yr⁻¹ MCWD) South-East region in our analysis is likely to be more widespread and severe (Figures 1-3). Moreover, there has been a slow shift to more dry-affiliated Amazonian tree genera⁵⁸, which have a lower biomass and are more savannah-like in nature⁵⁹ as some species reach their adaptive limits to ongoing drier conditions⁶⁰. Such a shift would threaten the permanence of the carbon sequestration potential of secondary forests as we have calculated in this study, especially if changes in tree communities lag substantially behind climatic changes^{17,58}.

11 - Line 251-253: make sure this message is in the summary and press release!

Response: We agree. This message needed to be made more clearly in the abstract. We have amended the last sentence of the abstract to now read:

“Implementing legal mechanisms to protect and expand secondary forests whilst supporting old-growth conservation is, therefore, key to realising their potential as a nature-based climate solution.”

12 - Line 262: also see this study using MapBiomas data for mapping natural regeneration in Atlantic Forest in Brazil:

Crouzeilles, R., H. L. Beyer, L. M. Monteiro, R. Feltran-Barbieri, A. C. Pessôa, F. S. Barros, D. B. Lindenmayer, E. D. Lino, C. E. Grelle, and R. L. Chazdon. 2020. Achieving cost-effective landscape-scale forest restoration through targeted natural regeneration. *Conservation Letters*:e12709.

Response: Thank you for directing us to this new paper using MapBiomas in the Atlantic Forest and have now cited this work. The sentence now reads:

“We follow a very similar methodology applied by recent studies^{10,18,48,49} to identify areas of secondary forest...”

13 - Line 269: could data from previous studies be used to estimate this?

Fearnside, P. M., and W. M. Guimaraes. 1996. Carbon uptake by secondary forests in Brazilian Amazonia. *Forest Ecology and Management* 80:35-46.

Helmer, E. H., M. A. Lefsky, and D. A. Roberts. 2009. Biomass accumulation rates of Amazonian secondary forest and biomass of old-growth forests from Landsat time series and the Geoscience Laser Altimeter System. *Journal of Applied Remote Sensing* 3:1-31.

Response: Indeed, this is an approach that we could have taken. However, we wanted to build an approach that has the potential to be used globally using remote sensing products in regions where field/local-scale data may be lacking. We wanted to (a) test and (b) highlight the potential of such an approach to be used in REDD+ and by others. We did compare our models with models by Fearnside and Guimaraes (see Supplementary Table 11).

14 - Line 295: did you examine the relationships among these variables (spatial co-variation)?

Response: In our original submission we did not examine this, but as stated in response 5 we believe we have now assessed this point in full, both the methods section and discussion. Given that we applied a cluster analysis to create regions of Amazonia based on similarities of the individual climate variables, we anticipate the variables to be spatially auto-correlated. This could also be assessed using a covariogram. As we carried out a cluster analysis, we deemed this approach to result in a very similar outcome as another approach such as a covariogram analysis. We will consider this approach in future research.

In the methods section an additional paragraph has been added, as well as Supplementary Note 5 to address the issue of spatial co-variation between the variables:

“Additionally, the variable importance assessed using the conditional random forest model better reflects the true impact of each variables regardless of any correlation between the variables compared to more traditional random forest models^{57,58}. In any geospatial analysis such as in this study, the variables are likely to be spatially correlated or have some degree of multicollinearity. It was therefore important to use a model that is not biased towards correlated variables.”

15 - Line 316: How do OG forests vary geographically in their aboveground carbon density?

Response: We have added a link to the Supplementary Figure 6, Supplementary Table 8 and 9 - which all show the varying AGC of old-growth forest across the regions:

We assumed that after a given amount of time, the AGC could return to levels equivalent to old-growth forests, and reach a precalculated asymptote. As such, we extracted the median, bias-corrected AGC value of old-growth forests under each variable condition from the ESA-CCI AGC product to represent the value of the asymptote (Supplementary Fig. 6 and Supplementary Tables 8 and 9).

16 - Line 352-355: This is similar to the approach used by Chazdon et al. (2016) (reference 14)

Response: Indeed it is. Thank you, we have referenced it in this section of the methods, the sentence now reads:

“Finally, we applied a similar approach as Chazdon et al. ¹⁴ and aged the standing SF in 2017 to model the carbon stock and annual carbon accumulation for the next decade...”

REVIEWERS' COMMENTS

Reviewer #2 (Remarks to the Author):

Your manuscript "Large carbon sink potential of Secondary Forests in the Brazilian Amazon to mitigate climate change" has been read by me and I think the text has been improved and the issues that were pointed solved. I think the study asked important questions, conducted rigorous measurements, analyses and statistical synthesis, and drew conclusions that are well supported by the data and have important implications. The manuscript represents a solid piece of work that should be published.

Reviewer #3 (Remarks to the Author):

Thank you for the careful revisions of this manuscript. All of my comments have been addressed and I am impressed with the quality and relevance of your study.

Below we copy and paste the comments from the two reviewers. We believe that we do not need to provide further changes to the manuscript with regards to their comments. We thank both reviewers for their comments throughout this process.

Kind regards,
Viola Heinrich on behalf of all co-authors (28/01/2021)

Reviewer #2 (Remarks to the Author):

Your manuscript "Large carbon sink potential of Secondary Forests in the Brazilian Amazon to mitigate climate change" has been read by me and I think the text has been improved and the issues that were pointed solved. I think the study asked important questions, conducted rigorous measurements, analyses and statistical synthesis, and drew conclusions that are well supported by the data and have important implications. The manuscript represents a solid piece of work that should be published.

Reviewer #3 (Remarks to the Author):

Thank you for the careful revisions of this manuscript. All of my comments have been addressed and I am impressed with the quality and relevance of your study.